# Molecular basis for the disruption of Keap1–Nrf2 interaction via Hinge & Latch mechanism

Yuta Horie[1,2,12], Takafumi Suzuki [1,12], Jin Inoue[3,4,12], Tatsuro Iso[1], Geoffrey Wells [5], Terry W. Moore[6], Tsunehiro Mizushima[7], Albena T. Dinkova-Kostova[8,9], Takuma Kasai [10,11], Takashi Kamei[2], Seizo Koshiba[3,4 ✉] & Masayuki Yamamoto [1,3,4 ✉]

The Keap1-Nrf2 system is central for mammalian cytoprotection against various stresses and a drug target for disease prevention and treatment. One model for the molecular mechanisms leading to Nrf2 activation is the Hinge-Latch model, where the DLGex-binding motif of Nrf2 dissociates from Keap1 as a latch, while the ETGE motif remains attached to Keap1 as a hinge. To overcome the technical difficulties in examining the binding status of the two motifs during protein-protein interaction (PPI) simultaneously, we utilized NMR spectroscopy titration experiments. Our results revealed that latch dissociation is triggered by low-molecular-weight Keap1-Nrf2 PPI inhibitors and occurs during p62-mediated Nrf2 activation, but not by electrophilic Nrf2 inducers. This study demonstrates that Keap1 utilizes a unique Hinge-Latch mechanism for Nrf2 activation upon challenge by non-electrophilic PPI-inhibiting stimuli, and provides critical insight for the pharmacological development of next-generation Nrf2 activators targeting the Keap1-Nrf2 PPI.

[1] Department of Medical Biochemistry, Tohoku University Graduate School of Medicine, Sendai, Japan. [2] Department of Surgery, Tohoku University Graduate School of Medicine, Sendai, Japan. [3] Tohoku Medical Megabank Organization, Tohoku University, Sendai, Japan. [4] The Advanced Research Center for Innovations in Next-Generation Medicine (INGEM), Tohoku University, Sendai, Japan. [5] UCL School of Pharmacy, University College London, London, UK. [6] Department of Pharmaceutical Sciences, College of Pharmacy, University of Illinois Cancer Center, University of Illinois at Chicago, Chicago, IL, United States. [7] Picobiology Institute, Graduate School of Life Science, University of Hyogo, Hyogo, Japan. [8] Jacqui Wood Cancer Centre, Division of Cellular Medicine, School of Medicine, University of Dundee, Dundee, United Kingdom. [9] Department Pharmacology and Molecular Sciences and Department of Medicine, Johns Hopkins University School of Medicine, Baltimore, MD, USA. [10] Laboratory for Cellular Structural Biology, RIKEN Center for Biosystems Dynamics Research, Yokohama, Japan. [11] PRESTO, JST, Kawaguchi, Japan. [12] These authors contributed equally: Yuta Horie, Takafumi Suzuki, Jin Inoue. ✉email: koshiba@megabank.tohoku.ac.jp; masiyamamoto@med.tohoku.ac.jp

The transcription factor Nrf2 (nuclear factor erythroid 2-related factor 2) plays a central role in cytoprotection against electrophilic and oxidative stress[1]. Under basal conditions, the Nrf2 protein is maintained at relatively low concentrations, as Nrf2 is constitutively ubiquitinated by Keap1 (Kelch-like ECH-associated-protein 1), an adaptor component of the Cul3 (Cullin 3)-based ubiquitin E3 ligase complex, and targeted for proteasomal degradation[2,3]. Upon exposure to electrophiles or oxidants, Nrf2 ubiquitination ceases, leading to its stabilization and nuclear translocation/accumulation, followed by inducible expression of Nrf2 target genes[4–6].

The Neh2 domain of Nrf2 is located at the N-terminus of the transcription factor and harbors two Keap1-binding motifs, a low-affinity DLGex motif and a high-affinity ETGE motif[7,8]. Whereas the DLGex motif possesses a three-helix structure and binds weakly to Keap1-DC (Double glycine repeat or Kelch, plus C-terminal) domain, the ETGE motif is a single β-hairpin structure that binds tightly to a pocket in Keap1-DC domain in a key-and-lock manner[8,9]. The binding of these two motifs of Nrf2 to the Keap1 homodimer (i.e., two-site binding) is strictly required for ubiquitination and proteasomal degradation of Nrf2[7,10]. Analytical centrifugation analyses support this notion, showing that the stoichiometry of the Keap1–Nrf2 complex is 2:1[11]. The two-site binding mode is supported by the finding that somatic mutations in various cancer cells occur with very high-frequency in the DLGex and ETGE motifs of Nrf2[12,13]. These mutations disrupt the two-site binding of Keap1–Nrf2 and lead to constitutive accumulation of Nrf2, supporting malignant growth of cancer cells[14].

Considering the difference in binding affinity between the DLGex and ETGE motifs and the necessity for the two-site binding for the Keap1-mediated ubiquitination of Nrf2, we hypothesized that the two-site binding is the key regulatory nexus of the Keap1–Nrf2 system. Based on a number of critical observations related to the two-site binding model of the Keap1–Nrf2 interaction, we have proposed the Hinge-Latch model[15,16]. For instance, immunoprecipitation or pull-down experiments have not detected dissociation between Keap1 and Nrf2 after exposure to Nrf2-activating compounds[3,17–21]. In the Hinge-Latch model, the Keap1–DLGex interaction first dissociates as a latch, while the Keap1-ETGE remains bound as a hinge regardless of the presence of Nrf2-activating stimuli[7,8,15–17].

A multitude of Nrf2-activating compounds have been reported, most of which are electrophilic and readily react with cysteine–thiols in Keap1[19,21]. Several cysteine sensors for electrophilic Nrf2 activators have been identified in Keap1[22–25]. Modifications of reactive cysteine residues in Keap1 have been demonstrated using mass spectrometry analyses[19,21,26–30]. Cys151 in the BTB (Broad-complex, Tramtrack and Bric-a-Brac) domain of Keap1 is the sensor for many Nrf2 activators, including 1-[2-cyano-3,12-dioxooleana-1,9(11)-dien-28-oyl] imidazole (CDDO-Im) and sulforaphane (SFN)[23,24]. Another functional sensor is Cys288; it is located in the IVR (Intervening Region) of Keap1, and is modified by 15d-PGJ2 (15-deoxy-$\Delta^{12,14}$-prostaglandin J2)[24]. Although modification of Cys151 is considered to disrupt Keap1–Cul3 interaction[11], modification of Cys288 is speculated to induce a conformational distortion to the Keap1-DC domain owing to the proximity between IVR and DC domains, subsequently leading to dissociation of Keap1–DLGex interaction (i.e., utilizing the Hinge-Latch mechanism).

In addition to electrophilic Nrf2-activating compounds, disruptors of the Keap1–Nrf2 protein-protein interactions (PPI) have been attracting increasing attention. For instance, p62/Sequestosome-1 (hereafter, referred to as p62) accumulates in autophagy-deficient hepatocytes and inhibits the Keap1–Nrf2 interactions[31]. The KIR (Keap1 Interaction Region) of p62[32] harbors an STGE motif, which acquires a high affinity for the Keap1-DC domain through phosphorylation of the serine residue[33]. In addition to p62, pharmacological non-electrophilic Keap1–Nrf2 PPI inhibitors have been developed. For instance, aryl bicyclic sulfonamide-based Keap1–Nrf2 PPI inhibitors such as PRL295[34] and NG262[35] have been shown to stabilize Nrf2 protein in cells. However, it is unknown how these Keap1–Nrf2 PPI inhibitors affect Keap1–DLGex binding and Keap1-ETGE binding.

In experiments using titration nuclear magnetic resonance (NMR) spectroscopy, the NMR signals corresponding to the intrinsically disordered Neh2 domain (comprising 98 amino acids) of Nrf2 experience line-broadening upon binding to the large molecular weight protein Keap1. Conversely, upon release from Keap1, NMR signals corresponding to the Neh2 domain recover by narrowing of the linewidth. Using this approach, we previously demonstrated the interaction between the Keap1-DC domain with the Nrf2-Neh2 domain and their molecular characteristics[7]. However, as we had utilized the DC domain protein that lacked the dimer-forming BTB domain of Keap1, and our amino-acid signal assignments were relatively limited, therefore we could not test the Hinge-Latch mechanism at that time.

To test the Hinge-Latch model, it is essential to examine the status of Keap1–DLGex binding and Keap1-ETGE binding simultaneously. However, concomitant measures of this two-site binding using conventional methodologies are technically difficult. Therefore, in this study, we developed a titration NMR approach. We also established a system in which full-length Keap1 homodimer interacts with isotope-labeled Neh2 protein and designed a fine detection system for signal recovery from individual interacting sites. We found that pharmacological Keap1–Nrf2 PPI inhibitors, NG262 and PRL295, disrupt the Keap1–DLGex binding preferentially to that of the Keap1-ETGE binding. Similarly, a phosphorylated p62 peptide also disrupts the Keap1–DLGex binding. In contrast, the electrophilic Nrf2-activating compounds CDDO-Im, SFN, and 15d-PGJ2 do not disrupt the Keap1–DLGex or the Keap1-ETGE interactions. Our results unequivocally demonstrate that Keap1–Nrf2 PPI inhibitors and p62, but not electrophilic cysteine-targeting compounds, utilize the Hinge-Latch mechanism for Nrf2 activation.

## Results

**NMR titration approach for two-site PPI assessment.** To ascertain whether the Hinge-Latch mechanism operates, it is essential to be able to verify the status of Keap1–DLGex binding and Keap1-ETGE binding independently. However, our previous attempts were hampered by the technical difficulty of validating the two-site binding event within one protein complex that comprises three subunits, i.e., two Keap1 proteins (in the context of the Keap1 homodimer) binding individually to two sites in one Nrf2 protein (Fig. 1a, left). In this study, we reasoned that titration NMR spectroscopy could provide a powerful means for observing the binding of the two sites independently and overcome this problem.

As shown in Fig. 1b, the Neh2 domain (98 amino acids) of Nrf2 is intrinsically disordered and, upon binding to the large molecular weight protein Keap1 dimer (molecular weight 140 kDa), NMR signals corresponding to the Neh2 domain experience line-broadening. The line-broadening likely accounts for their interaction rather than a conformational exchange, because co-crystal structures of Keap1–DLGex and Keap1-ETGE have shown that the same pocket of Keap1 binds to DLGex and ETGE[8,9]. On the other hand, when released from the Keap1 homodimer, NMR signals corresponding to the binding sites of

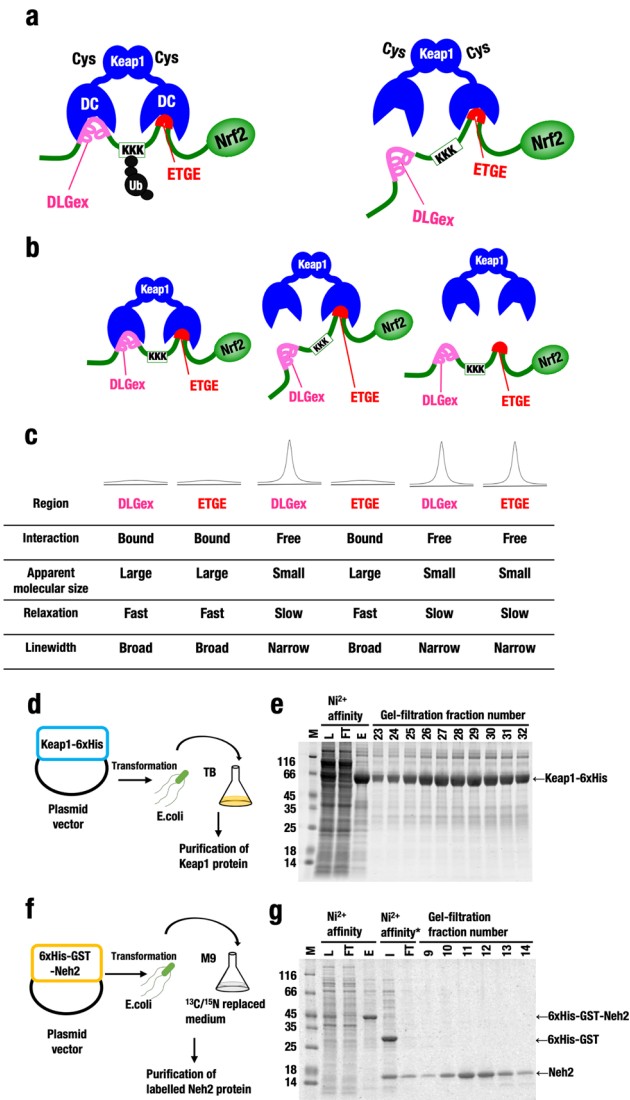

**Fig. 1 Verification of the Hinge-Latch model by NMR titration method.**
**a** Working hypothesis for the Hinge-Latch model of activation of Nrf2 by electrophilic and non-electrophilic inducers. This model was proposed based on the two-site binding model; DLGex and ETGE motifs of Nrf2 bind individually to the similar pocket structure in the Keap1-DC domain of the Keap1 homodimer. According to the Hinge-Latch model, DLGex site dissociation alone sufficiently disturbs the ubiquitin ligase activity of the Keap1 complex[14]. **b** Hypothetical statuses of the Keap1–Nrf2 complex. As ETGE motif has a 200-fold higher binding affinity to Keap1-DC domain than DLGex motif does, three statuses of the Keap1–Nrf2 protein complex can be postulated; two sites bound (left), only ETGE site bound (middle), and fully dissociated (right). **c** Rationale of the NMR titration technique. A schematic depiction of a 1D NMR peak is shown. The Neh2 domain of Nrf2 (98 amino acids) is intrinsically disordered and, upon binding to the large molecular weight protein Keap1 dimer (molecular weight ~140 kDa), the NMR signals corresponding to the Neh2 domain experience line-broadening. On the other hand, when released from Keap1 homodimer, the NMR signals corresponding to the binding sites of Neh2 domain recover by narrowing of linewidth. The NMR titration experiment enables us to detect the binding of each individual site and to validate the three conditions shown in **b**. **d** Scheme for induction and expression of recombinant full-length Keap1-6xHis protein. **e** Purification processes of recombinant full-length Keap1-6xHis protein. **f** Scheme for induction and expression of recombinant 6xHis-GST-$^{13}$C$^{15}$N Neh2 protein. **g** Purification processes of recombinant 6xHis-GST-$^{13}$C$^{15}$N Neh2 protein. The protein was purified by Ni$^{2+}$ affinity column chromatography and gel filtration chromatography. For the preparation of Neh2 domain, after Ni$^{2+}$ affinity column chromatography, 6xHis-GST tag was cleaved by turbo TEV protease and removed by second Ni$^{2+}$ affinity chromatography (*). *M* molecular marker, *L* lysate, *FT* flow through, *I* input, *E* eluate.

the Neh2 domain recover, shown by narrowing of the linewidth (Fig. 1a, right). In fact, we previously demonstrated that the Keap1-DC domain (molecular weight ~30 kDa for the monomer) interacts with the Neh2 region of Nrf2 by NMR titration experiments[7].

In this study, therefore, we have addressed this issue by utilizing titration NMR spectroscopy, as the unique mode of interaction between Keap1 and Nrf2 allows us to apply this technique for the two-site PPI analysis. For this purpose, we first labeled the Neh2 domain of Nrf2 with $^{13}$C and $^{15}$N during its expression and generated the Keap1 dimer–Neh2 complex (Fig. 1b, left). As this complex did not give rise to any clear-cut signal, we then titrated the complex with various components to observe the appearance of a signal. Based on the Hinge-Latch model, we surmised the presence of the ETGE single-binding condition (Fig. 1b, middle) and the DLGex and ETGE double-dissociated condition (right). We expected to detect binding of each individual site and to validate these three conditions by NMR titration (Fig. 1c).

**Full-length Keap1 and labeled Neh2 domain structure.** To conduct NMR structure analyses, in the previous study, we used a truncated Keap1-DC protein, which lacks the N-terminal region containing the BTB domain[7]. The N-terminal region is required for the Keap1–Keap1 protein interaction and the formation of

Keap1 homodimer[36]. As the lack of N-terminus may affect binding of Keap1 to the DLGex and ETGE motifs of Nrf2, we decided to use full-length Keap1 to validate the Neh2 domain structure and the two-site-binding Hinge-Latch model. To this end, we first expressed full-length mouse wild-type Keap1-6×His in bacteria by the transformation of an expression plasmid vector, and performed purification by Ni$^{2+}$ affinity column chromatography, anion exchange chromatography, and gel filtration chromatography (Fig. 1d). This approach resulted in the preparation of highly purified recombinant full-length Keap1 protein (Fig. 1e).

To prepare the Neh2 protein for the NMR experiment, 6xHis-GST tagged Neh2 protein was expressed in bacteria transformed with plasmid vector. The 6xHis-GST tagged Neh2 protein was uniformly labeled by culturing in M9 minimal medium in which carbon and nitrogen were replaced with stable isotopes $^{13}$C and $^{15}$N (Fig. 1f). After Ni$^{2+}$ affinity column chromatography, the 6xHis-GST tag was cleaved by turbo TEV protease and removed by a second Ni$^{2+}$ affinity chromatography step (Fig. 1g). Thus, we also prepared highly purified stable isotope-labeled recombinant Neh2 domain protein.

**Assignment of the sequence-specific backbone of Neh2.** Utilizing purified labeled Neh2 protein, we measured the 2D transverse relaxation optimized spectroscopy (TROSY)-HSQC NMR spectrum to detect the HN peaks from the main chain of amino acids of the labeled Neh2 domain protein. As previously reported[7], the TROSY-HSQC spectrum of unbound Neh2 exhibited a limited chemical shift dispersion (Fig. 2a). Backbone amide protons resonate between 7.7 and 8.7 ppm around the random coil region of 8.0–8.5 ppm[37], a feature typical of intrinsically disordered proteins[38]. Except for overlapped resonances, we were able to make the sequence-specific backbone assignments for Neh2. Assigned amino acids are indicated in bold in Fig. 2b.

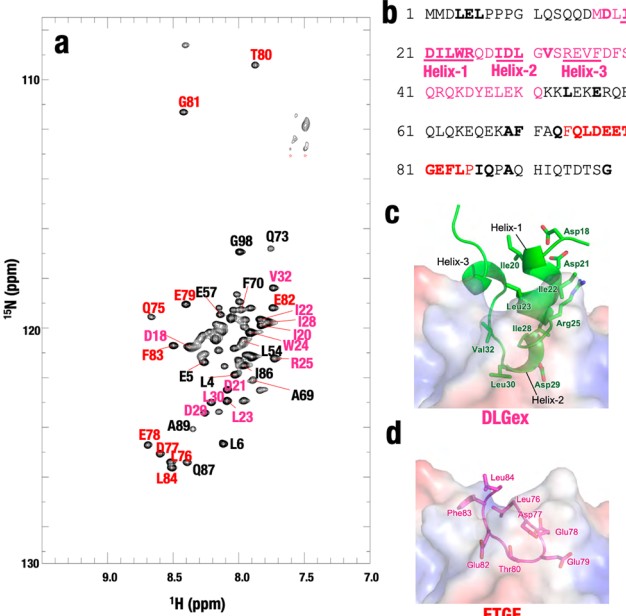

**b**
```
1   MMD**LEL**PPPG LQSQQD**MDLI**
21  **DILWR**QD**IDL** **GV**SREVFDFS
    Helix-1  Helix-2  Helix-3
41  QRQKDYELEK QKK**LEK**E**RQE**
61  QLQKEQE**AF** FAQ**FQLDEET**
81  **GEFLP**IQP**AQ** HIQTDTS**G**
```

**c** DLGex

**d** ETGE

Fig. 2 Assignment of HN peaks of amino-acid main chain of labeled Neh2.
**a** 2D TROSY-HSQC spectrum shows HN peaks of the main chain of amino acids of labeled Neh2 domain protein. **b** Alignment of mouse Neh2 domain. The assigned amino acids are in bold. The amino acids in Helix-1, Helix-2, and Helix-3 in the DLGex motif are underlined. **c, d** Structure of DLGex (**c**) and ETGE (**d**) motifs associating with the Keap1 pocket. Note that the DLGex motif possesses three-helix regions (Helix-1, 19–25; Helix-2, 28–30; Helix-3, 34–37) (PDB 3WN7, ref. [8]), whereas the ETGE motif forms a β-hairpin structure (PDB 2DYH, ref. [16]). We have proposed the two-site binding model in which each motif binds to a similar pocket structure in the Keap1-DC domain of the Keap1 homodimer (as shown in Fig. 1a).

Unlike the ETGE motif that forms a β-hairpin structure and binds tightly to the Keap1-DC domain[9], the DLGex motif forms three helical regions (Helix-1, 19–25; Helix-2, 28–30; Helix-3, 34–37) in the Keap1-bound form (Fig. 2c) (PDB 3WN7)[8]. We assigned amino acids corresponding to Helix-1 (ILe20, Asp21, Ile22, Leu23, Trp24, and Arg25) and Helix-2 (Ile28, Asp29, and Leu30) in the DLGex motif, whereas amino acids corresponding to Helix-3 could not be assigned. Amino acids corresponding to ETGE motif (Gln75, Leu76, Asp77, Glu78, Glu79, Thr80, Gly81, Glu82, Phe83, and Leu84) were also assigned (Fig. 2d). Keap1 dimer is unstable at pH <8.0 at protein concentrations suitable for the sequence-specific backbone assignment. Despite efforts for years, we still could not accomplish the full backbone assignment. Nonetheless, in this study, we have succeeded substantial extension of the assignment from our previous assessment of the Neh2 domain NMR structure[7] and this enabled us to execute informative NMR titration experiments.

**Full-length Keap1 binds to Neh2 via DLGex and ETGE motifs in a 2:1 stoichiometry.** We performed NMR titration analyses to map the two binding sites in the Neh2 domain of Nrf2 with the full-length Keap1 homodimer. In accordance with our expectation, we found that there was an increasing disappearance of the resonance signals corresponding to the DLGex and ETGE motifs in the 2D TROSY-HSQC spectra upon complex formation with full-length Keap1 (Fig. 3a–c). The loss of NMR signals is most likely owing to an increase in the apparent molecular weight caused by the protein complex formation. We titrated 50-μM labeled Neh2 domain protein with either an equivalent amount or double the amount of Keap1 protein (Fig. 3b, c, respectively).

In the bar graph representations of signal peak intensity, we calculated the signal peak intensity under each condition by setting the signal peak observed with the unbound Neh2 domain protein to 100% (Fig. 3d–f). Although the relative intensity of Helix-1 in DLGex motif was reduced to ~40% by the addition of an equal amount of Keap1 protein, the relative intensity of Helix-2 in the DLGex motif was reduced to ~70% (Fig. 3e), suggesting that Helix-1 interacts with Keap1 preferentially and more tightly compared with Helix-2. The relative intensity of the ETGE motif was reduced to ~20–30%, indicating that ETGE motif interacts with Keap1 much more tightly than the DLGex motif. These results are consistent with previous reports that the ETGE motif of Nrf2 has a higher binding affinity to Keap1 than the DLGex motif when Keap1-DC domain lacking the BTB domain-mediated homodimerization was used[7].

The signals corresponding to both helices disappeared almost completely on the addition of twofold excess amount of Keap1 protein (Fig. 3f). Of note, the signals from the ETGE motif also disappeared completely in this condition, indicating the formation of two-site-binding between the Keap1 homodimer and both DLGex and ETGE motifs in Neh2. These results strongly support the Hinge-Latch model in which Keap1 homodimer binds to the Neh2 domain of Nrf2 with 2:1 stoichiometry via the lower-affinity DLGex motif and the higher affinity ETGE motif.

Closer inspection of the signal peak intensity revealed that there were still signals from the N-terminus (Leu4, Glu5, and Leu6) and C-terminus (Gly98) of Nrf2 (Fig. 3d–f), indicating that the N- and C-terminal regions of the Neh2 domain of Nrf2 do not interact with Keap1 under these assay conditions, resulting in slow relaxation. By contrast, the signals corresponding to the intervening region between DLGex and ETGE motifs (Leu54, Glu57, Ala69, Phe70, and Gln73) had disappeared, implying that this region also associates with Keap1 following the interaction via DLGex and ETGE motifs.

**Keap1–Nrf2 PPI inhibitors inhibit the Keap1–DLGex interaction preferentially.** The Keap1–Nrf2 PPI inhibitors are very attractive next-generation Nrf2 activators, because PPI inhibitors are more specific and less toxic than thiol-modifying chemicals[39]. The Keap1–Nrf2 PPI inhibitors are designed to bind the Keap1-DC pocket, and thereby interfere with Nrf2 binding. However, the precise molecular details of how PPI inhibitors affect the interactions between Nrf2 and Keap1 remain elusive. To address this issue, we examined the effects of low-molecular-weight Keap1–Nrf2 PPI inhibitors, PRL295[34] and NG262[35], both of which bind to the Keap1-DC domain and activate Nrf2.

PRL295 binds to the pocket located in the Keap1-DC β-propeller structure (PDB 6UF0). NG262 also binds to the same pocket. This structural examination suggests that both PRL295 and NG262 compete with the binding motifs in the Neh2 domain. Therefore, we hypothesized that the molecular mechanism of action of these compounds can be explained by the Hinge-Latch mechanism. Specifically, we postulated a scenario in which the DLGex–Keap1 interaction is dissociated first by these compounds, whereas the Hinge site may also dissociate if the compounds are in vast excess.

To test this hypothesis, we conducted a new series of NMR titration experiments. We titrated the Keap1–Neh2 interaction (full interaction; Keap1:Neh2 ratio = 2:1) by using PRL295 or NG262. We first mixed 50-μM-labeled Neh2 domain (Supplementary Fig. 1a) with 100-μM Keap1 protein and generated the Keap1–Neh2 fully interacting complex (Supplementary Fig. 1b). We then titrated this protein complex with increasing amounts of PRL295 or NG262. We found that PRL295 nicely inhibited the Keap1–Neh2 interaction even at an equimolar concentration

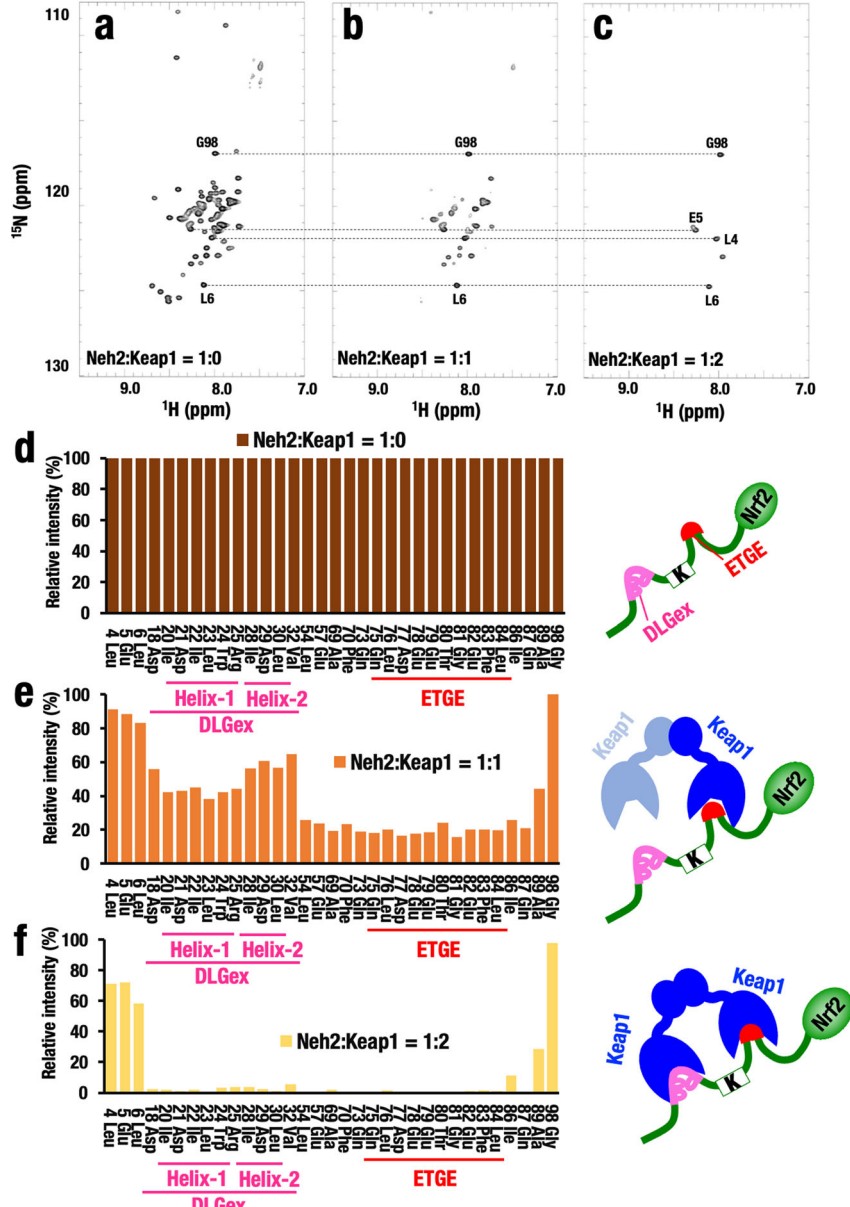

**Fig. 3 Keap1 binds to Neh2 domain of Nrf2 via DLGex and ETGE motifs at 2:1 stoichiometry. a–c** 2D TROSY-HSQC spectrum shows HN peaks of the main chain of amino acids of labeled Neh2 domain protein in the absence of Keap1 (**a**), in the presence of equal amount of Keap1 (**b**) or twofold excess of Keap1 (**c**). **d–f** Relative peak intensity of the NMR signal of Neh2 only (**d**), Neh2:Keap1 = 1:1 (**e**) or Neh2:Keap1 = 1:2 (**f**). Relative peak intensity of the NMR signal of each amino-acid residue is calculated after the addition of Keap1, setting the peak intensity of Neh2 only as 100%. Note that we quantitated it by the concentration of the individual Keap1 protomers rather than homodimer here.

(Supplementary Fig. 1c) and fivefold excess of the compound fully inhibited the interaction (Supplementary Fig. 1d). NG262 also showed a similar titration profile, thus, the Keap1–Neh2 interaction dissociated with equimolar concentration of NG262 (Supplementary Fig. 1e) and fivefold excess fully inhibited the interaction (Supplementary Fig. 1f). These results thus demonstrate that the PPI inhibitors PRL295 and NG262 dissociate the Keap1–Neh2 interaction.

Detailed inspection of these titration experiments revealed marked difference between the DLGex–Keap1 and ETGE–Keap1 bindings. It should be noted that the NMR signals corresponding to the DLGex motifs reached almost half of the full signals by the addition of equal amounts of PRL295 (Fig. 4a) or NG262 (Fig. 4b). The DLGex motif signal increased to an almost full (non-complexed) signal level by the addition of twofold excess

PRL295 or NG262. In this case, the ETGE–Keap1-binding signals increased to approximately half of the full signals. These results unequivocally demonstrate that both compounds inhibit the DLGex–Keap1 binding preferentially compared with ETGE–Keap1 binding, strongly supporting the contention that these PPI inhibitors act through the Hinge-Latch mechanism.

This conclusion is further supported by the merge images of PRL295-DLGex and NG262-DLGex (Supplementary Fig. 2a and b, respectively). The binding profiles of these compounds are very similar and deeply localized in the pocket in the Keap1-DC structure. Closer inspection of the titration profiles further revealed that Helix-2 in the three helices of DLGex dissociated more quickly from the Keap1 pocket than Helix-1 did in the titration experiments by PRL295 and NG262 (Fig. 4a, b, respectively). Helix-3 could not be examined in this study. The

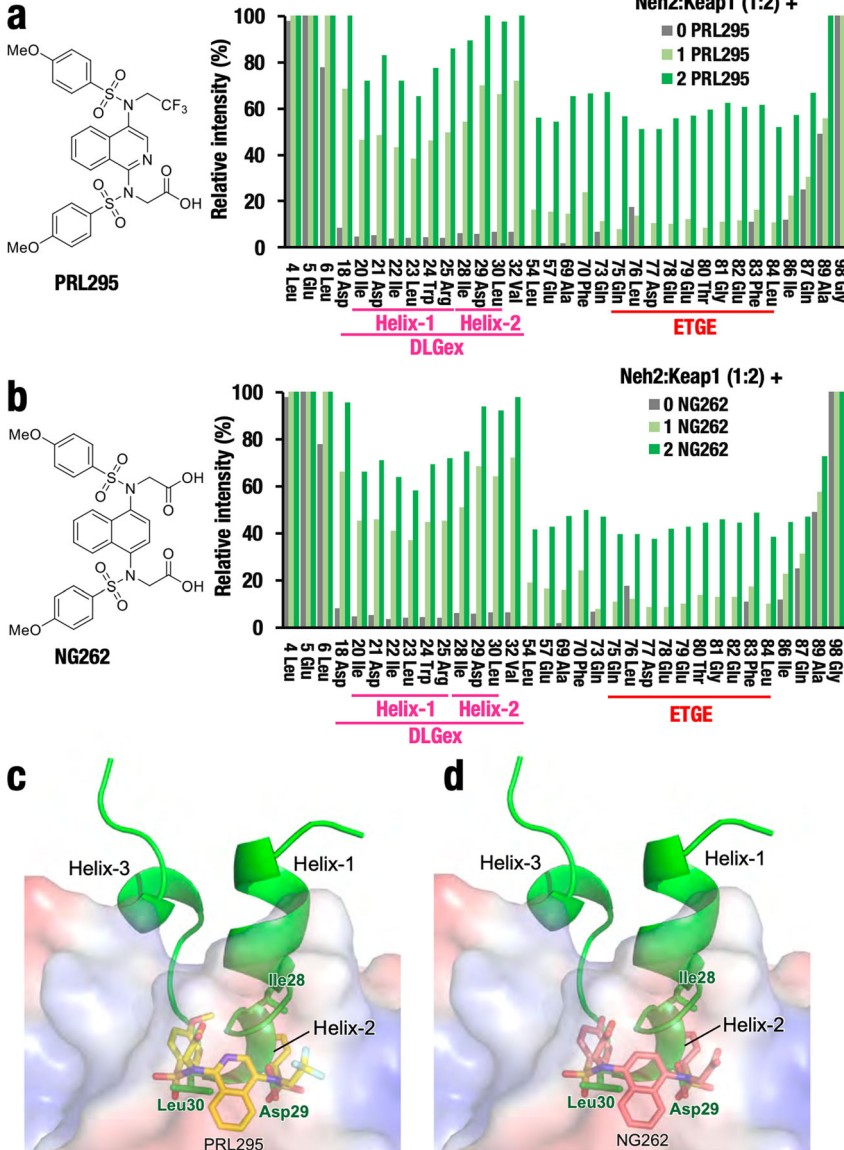

**Fig. 4 Keap1–Nrf2 PPI inhibitors preferentially inhibit the Keap1–DLGex interaction. a, b** NMR titration experiment of labeled Neh2–Keap1 complex (1:2) by the addition of equal amount or twofold excess of PPI inhibitor, PRL295 (**a**) and NG262 (**b**). The relative peak intensity of the NMR signal of each residue was calculated (set at 100% for Neh2 only). **c, d** Merged images of DLGex with PRL295 (**c**) or NG262 (**d**) associating with Keap1 pocket. Image of co-crystal Keap1-PRL295 complex (PDB 6UF0, ref. [34]) was used. Note that bindings of these compounds heavily overlap with that of DLGex Helix-2 (**c, d**) and these compounds titrate the binding of Helix-2 more efficiently than that of Helix-1 (**a, b**).

relative intensities corresponding to DLGex increase to a greater extent in response to these inhibitors. Inspection of the merge images of PRL295-DLGex and NG262-DLGex further revealed that the binding positions of both PPI inhibitors in the Keap1 pocket heavily overlapped with that of Helix-2 (Fig. 4c, d). These results indicate that PRL295 and NG262 preferentially inhibit the DLGex–Keap1 interaction via competing with Helix-2 of DLGex for binding to Keap1. These findings provide valuable information for the development of next-generation Nrf2 inducers that specifically inhibit the Keap1–Nrf2 interaction without electrophilic insults.

**Electrophilic Nrf2 inducers do not inhibit the Keap1–Neh2 interaction.** Electrophilic Nrf2 activators belong to several classes based on the utilization of specific sensor cysteine residues in Keap1[24,25,40]. The best characterized class of Nrf2 activators are Cys151-targeting compounds including CDDO-Im and SFN[23,24].

Therefore, we next tested the effect of CDDO-Im and SFN on the Keap1–Neh2 interaction.

We first mixed 50-µM labeled Neh2 domain protein (Supplementary Fig. 3a) with 100-µM Keap1 protein and generated the Keap1–Neh2 protein complex in which most of the NMR signals were disappeared (Supplementary Fig. 3b). We then titrated the protein complex with CDDO-Im (Fig. 5a). In contrast to the PPI inhibitors, addition of CDDO-Im at even the maximally soluble concentration (10-fold excess) did not increase the NMR signals corresponding to the DLGex or ETGE motifs (Supplementary Fig. 3c, d; Fig. 5a), indicating that CDDO-Im does not dissociate the Keap1–Neh2 interaction.

Similarly, addition of SFN at even the maximally soluble concentration (10-fold excess) did not increase the signals corresponding to the DLGex or ETGE motifs (Supplementary Fig. 3e, f; Fig. 5b), indicating that the SFN treatment did not dissociate the DLGex and ETGE motifs from Keap1. These results

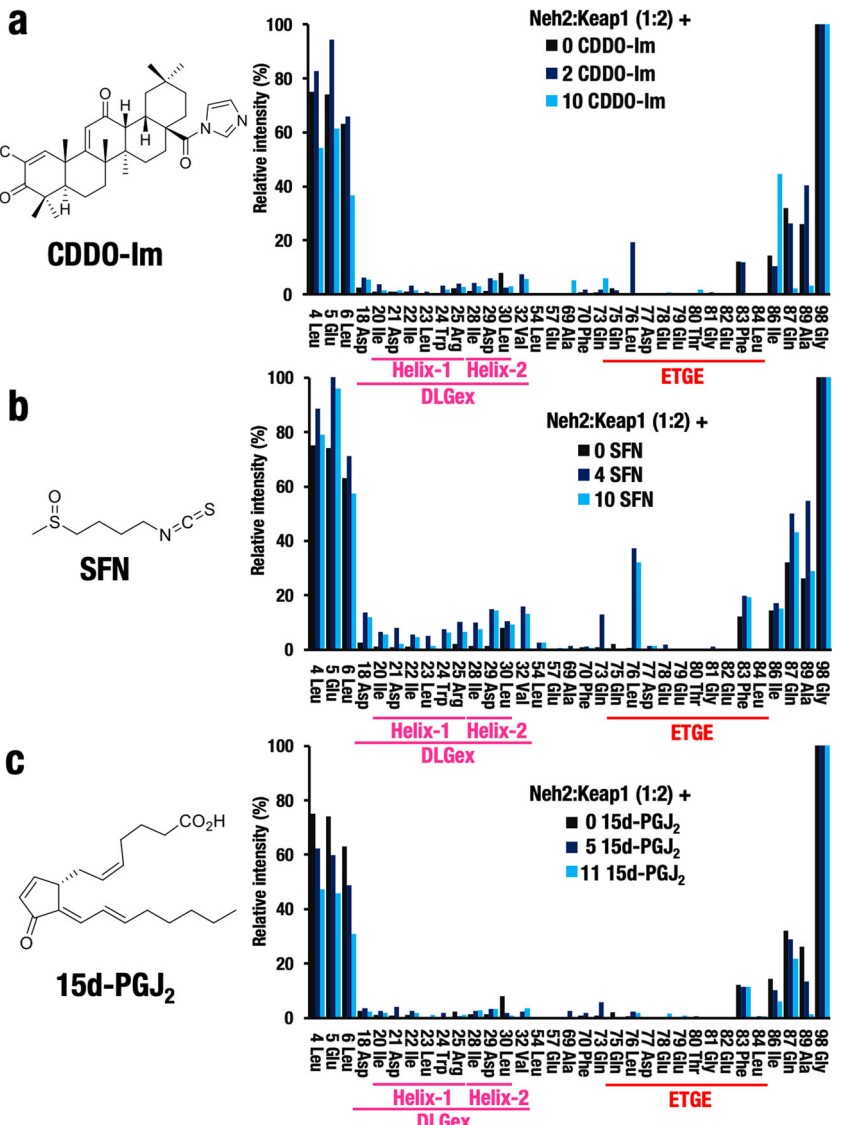

**Fig. 5 Electrophilic Nrf2 activators do not disrupt the Keap1–Neh2 interaction.** NMR titration experiment of labeled Neh2–Keap1 complex (1:2) with electrophilic Nrf2 inducers, CDDO-Im **a**, SFN **b**, and 15d-PGJ₂ **c**. Note that none of these compounds titrate the Neh2–Keap1 binding efficiently in the exploited range of concentrations. The relative peak intensity of the NMR signal of each amino-acid residue is calculated after setting Neh2 only as 100%.

demonstrate that CDDO-Im and SFN activate Nrf2 without disrupting the Keap1–Neh2 interaction, consistent with the notion that modification of Cys151 is considered not to disrupt the Keap1–Neh2 interaction but to disrupt the Keap1–Cul3 interaction instead[11]. Curiously, we also found that SFN increased the signals corresponding to Leu76, but the significance or the molecular basis for these signals remains to be clarified.

15d-PGJ₂ belongs to another class of electrophilic Nrf2 activators, those that modify Cys288 of the IVR domain in Keap1. As a modification of Cys288 is speculated to be conveyed as conformational distortion to the Keap1-DC domain owing to the proximity between the IVR and DC domains, we surmised that the modification of Cys288 by 15d-PGJ₂ might dissociate the Keap1–DLGex binding (i.e., utilize the Hinge-Latch mechanism). To test this hypothesis, we examined whether 15d-PGJ₂ disrupt Keap1–DLGex binding (Fig. 5c; Supplementary Fig. 3g, h). Unexpectedly, we found that 15d-PGJ₂ did not increase the signals of DLGex and ETGE motifs, indicating that 15d-PGJ₂ activates Nrf2 without affecting the Keap1–Neh2 interaction. Taken together, these results unequivocally demonstrate that representative classical electrophilic Nrf2 inducers, CDDO-Im,

SFN, and 15d-PGJ₂, which are known to modify cysteine sensors in Keap1, and consequently activate Nrf2, do not act through the Hinge-Latch mechanism.

**p62-KIR peptides preferentially inhibit the Keap1–DLGex interaction.** Next, we explored whether other Nrf2-inducing machineries that are known to affect the binding between Nrf2 and Keap1 in a physiological context employ the Hinge-Latch mechanism. To the best of our knowledge, the most characterized inducer in vivo that competitively binds to Keap1 is p62/Sequestosome-1, an adaptor/chaperone of cellular proteins guided to autophagy[31]. Our previous studies have shown that the KIR (Keap1-Interacting Region) of p62 contains an STGE sequence[32] and the phosphorylated STGE (pSTGE) sequence has a higher affinity for the Keap1-DC than non-phosphorylated STGE does[33]. Therefore, in order to test how p62 affects the Keap1–Neh2 interaction, we tested non-phosphorylated and phosphorylated KIR peptides using the titration NMR technique.

To this end, 50-μM labeled Neh2 domain protein was bound to 100-μM Keap1 protein at a molar ratio of 1:2 (Supplementary

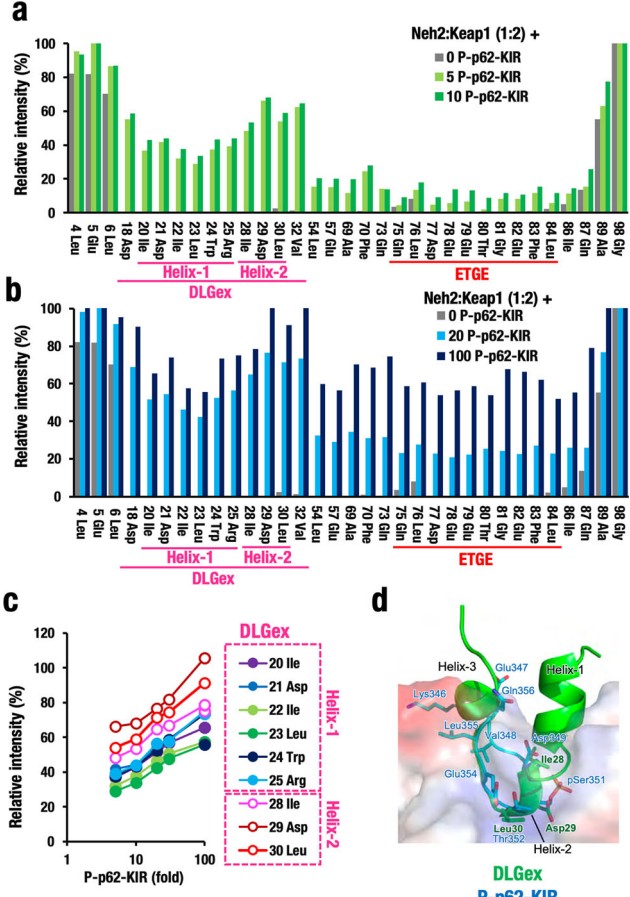

**Fig. 6 Phosphorylated p62-KIR peptide preferentially inhibits Keap1–DLGex interaction. a** Relative peak intensity of each amino-acid residue of Neh2 in the Neh2–Keap1 complex in the titration experiments by the addition of five- and tenfold excess of phosphorylated p62-KIR (P-p62-KIR) peptide. Relative peak intensity of the NMR signal of each residue is calculated setting Neh2 only as 100%. **b** Relative peak intensity of each amino-acid residue of Neh2 in the Neh2–Keap1 complex by the addition of 20- and 100-fold of P-p62-KIR peptide. **c** Relative peak intensity of the amino-acid residues within the DLGex motif after the addition of P-p62-KIR peptide. Closed and open circles indicate the amino-acid residues corresponding to Helix-1 and Helix-2, respectively. **d** Merged structural images of the Keap1-DC with the DLGex and P-p62-KIR. Note that binding of P-p62-KIR heavily overlaps with that of DLGex Helix-2 **d** and the P-p62-KIR titrates the binding of Helix-2 more efficiently than that of Helix-1 (**a**, **b**).

Fig. 4a), and then titration experiments were performed using the phosphorylated p62-KIR peptide (P-p62-KIR); the Keap1–Neh2 complex was titrated using a 5-, 10-, 20-, 30-, or 100-fold excess of P-p62-KIR peptide (Fig. 6a, b; Supplementary Fig. 4b, c). We found that the P-p62-KIR peptides preferentially increased the signals corresponding to the DLGex motif compared to the ETGE motif in a concentration-dependent manner. Thus, the signals for DLGex appeared even with a fivefold excess of P-p62-KIR (Fig. 6a; light green bars), whereas the signals for ETGE only started appearing when 20-fold excess was used (Fig. 6b; blue bars), and was saturated with a 100-fold excess (black bars) of P-p62-KIR. These data show that a 100-fold excess of P-p62-KIR peptide can compete with the ETGE motif for binding to Keap1. Of note, we found that Helix-2 in DLGex motif dissociated more quickly from the Keap1 pocket than Helix-1 did during the titration by P-p62-KIR (Fig. 6c), which is consistent with the observation using the Keap1–Nrf2 PPI inhibitors. These results

thus demonstrate that the P-p62-KIR peptide disrupted the binding of DLGex–Keap1 (Latch site; Fig. 6c) more efficiently than that of ETGE–Keap1 (Hinge site; Supplementary Fig. 4d) and the dissociation of the DLGex motif starts from Helix-2 (Fig. 6d).

We also examined whether non-phosphorylated p62-KIR (Non-P-p62-KIR) peptide affects the Keap1–Neh2 interaction. We tested 20- and 100-fold excess concentrations of the Non-P-KIR peptide, and found that this peptide, albeit weakly, gave rise to increase of the signals corresponding to the DLGex motif (Supplementary Fig. 5a, b). This Non-P-p62-KIR peptide could only suppress partially the ETGE–Keap1 binding (Supplementary Fig. 6c). Overall, the efficiency of DLGex dissociation by Non-P-p62-KIR is much lower than that by the P-p62-KIR peptide, in full agreement with the report that phosphorylation of p62-KIR increases markedly its binding affinity for Keap1[33].

These results strongly support the notion that the Hinge-Latch mechanism is functional in vivo. The pSTGE motif in the P-p62-KIR peptide interferes with binding of DLGex of Neh2 to Keap1 thus opening the Latch site, which suppresses efficient ubiquitination and rapid degradation of Nrf2. Taken together, this study supports the contention that the Hinge-Latch mechanism is operating in the Nrf2 activation caused by p62 accumulation induced by autophagy impairment as well as by pharmacological non-electrophilic Keap1–Nrf2 PPI inhibitors, but not classical electrophilic Nrf2 inducers.

## Discussion

This study addresses the long-lasting question of how electrophiles and non-electrophilic PPI inhibitors target Keap1 to activate Nrf2. The cysteine–thiol-based sensor activity of Keap1 has been examined extensively[20–25,40]. Similarly, various structure–function analyses of the Keap1–Nrf2 interaction have revealed the importance of the two-site binding between the Keap1 homodimer (i.e., two DC domains) and the Neh2 domain of Nrf2 (via DLGex and ETGE motifs)[7,8,10,15,16], and we proposed the Hinge-Latch model as a plausible mechanism for the Keap1-mediated Nrf2 activation[15,16]. This model is based on the fact that the DLGex and ETGE motifs show approximately two orders of magnitude difference in the binding affinity to the Keap1-DC domain[7]. Various lines of evidence including somatic mutation analyses in clinical cancer studies support the presence of this model[12,13]. However, the model has not been validated to date owing to technical difficulties in examining the DLGex–Keap1 binding and ETGE–Keap1 binding simultaneously. To overcome this problem, in this study we utilized NMR titration, which is capable of simultaneous and sequence-specific assignment of the DLGex and ETGE motifs. Utilizing this method, we found that the Hinge-Latch mechanism operates during the activation of Nrf2 by p62 accumulation (which can be induced by autophagy deficiency) as well as pharmacological Keap1–Nrf2 PPI inhibitors such as NG262 and PRL295, but not by electrophilic Nrf2 inducers such as CDDO-Im, SFN, or 15d-PGJ$_2$ (Fig. 7).

Moreover, our current study unequivocally demonstrates that pharmacological Keap1–Nrf2 PPI inhibitors, NG262 and PRL295, preferentially disrupt the DLGex–Keap1 binding compared with the ETGE–Keap1 binding, confirming the Hinge-Latch model. This fact strongly argues that these PPI disruptors preferentially interfere with the DLGex–Keap1 binding owing to the lower affinity of the DLGex motif for Keap1 when compared with the ETGE motif[7]. The DLGex motif possesses three-helix regions (Helix-1, 19–25; Helix-2, 28–30; Helix-3, 34–37)[8] and our current assignment of signals allowed us to detect the binding of Helix-1 and Helix-2. Notably, the Helix-2 regions (28–30) were much more profoundly affected by the PPI disruptors than the Helix-1

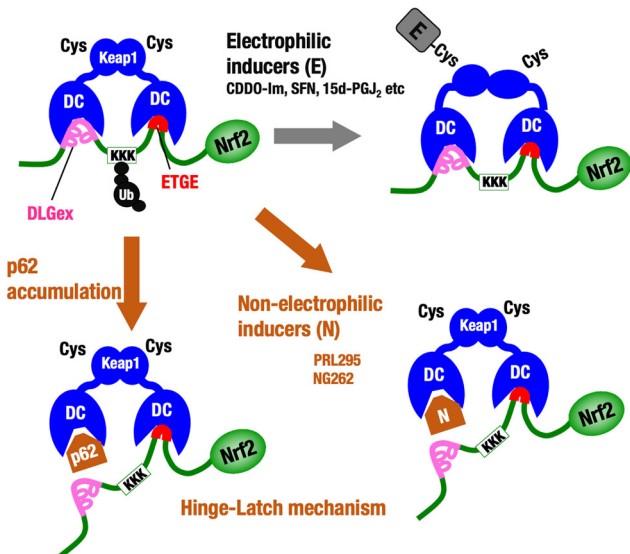

**Fig. 7 The Hinge-Latch model updated.** Pharmacological Keap1–Nrf2 PPI inhibitors disrupt the DLGex–Keap1 interaction preferentially to the ETGE–Keap1 interaction, conforming the concept of the Hinge-Latch model. Similarly, p62 accumulation disrupts the DLGex–Keap1 interaction preferentially and activates Nrf2, conforming to the Hinge-Latch model and supporting the notion that this mechanism operates in physiological as well as pathological contexts. In contrast, electrophilic Nrf2-activating compounds activate Nrf2 without disrupting the Neh2-Keap1 interaction, implying that these Keap1 thiol-modifying chemicals utilize mechanisms distinct from the Hinge-Latch mechanism.

region (19–25). This observation demonstrates that Helix-2 is much more sensitive to the PPI disruptors than Helix-1. This notion is supported by the protein structure assignments in which the binding of the PPI inhibitors nicely overlaps with the binding interface of Helix-2. We surmise that this information will be crucial for the design of novel Keap1–Nrf2 PPI inhibitors capable of efficient disruption of the DLGex–Keap1 interaction in the future development of next-generation Nrf2 inducers.

It has been shown that Keap1 is degraded through the autophagy pathway[41]. Specific protein degradation process through the autophagosome is referred to as selective autophagy[32]. Selective autophagy has an important role in physiological as well as pathological processes, including neurodegenerative diseases and oncogenesis[42]. The decline of autophagy is always accompanied by the accumulation of p62[43]. In this regard, many lines of experimental evidence have revealed that a high-level accumulation of p62 promotes the activation of Nrf2[32], but the molecular mechanism underlying the p62-mediated Nrf2 activation remained to be clarified. By use of NMR titration, this study demonstrates that the Hinge-Latch mechanism is the mechanism operating during Nrf2 activation by the p62 accumulation. Thus, the Hinge-Latch mechanism operates for cellular biological processes as well as PPI inhibitors.

Of note, although equimolar amounts of NG262 or PRL295 effectively disrupted the Keap1–Neh2 interaction, more than a 20-fold excess of the p62-KIR peptide was required for disruption of the Keap1–Neh2 interaction. We envisage that this difference reflects the presence of complex molecular interactions within the intracellular operation of the Hinge-Latch mechanism. One plausible explanation for the difference is to consider the p62 protein oligomerization via PD1 and UBA domains[44,45], which are absent in the p62-KIR peptide. The oligomerization of p62 is expected to enhance the local concentration of p62 protein and thus may effectively disrupt the DLGex–Keap1 interaction in

cells. Supporting this possibility, p62 is known to be an Nrf2 target gene[5], implying that positive feedback regulation by Nrf2 increases p62 levels in cells and helps to disrupt the DLGex–Keap1 interaction. An alternative explanation is that additional protein(s) assist the interaction of p62 and Keap1-DC domain in vivo[46,47], which are absent in our in vitro titration experiment.

Another important finding in this study is that the electrophilic Nrf2 activators, including CDDO-Im, SFN, and 15d-PGJ₂ do not affect the Keap1–Nrf2 interaction. As the sensor Cys151 for CDDO-Im and SFN is located in the BTB domain of Keap1, distant from the Latch site, it has been assumed that these two compounds utilize mechanisms other than the Hinge-Latch model. Consistent with this assumption, a Förster resonance energy transfer study in live cells expressing fluorescently-tagged Keap1 and Nrf2 and has shown that electrophilic inducers do not disrupt the Keap1–Nrf2 interaction[48]. In this regard, there remained a possibility that 15d-PGJ₂ may use the Hinge-Latch mechanism, because the sensor Cys288 for 15d-PGJ₂ is located in the IVR domain of Keap1, in a relatively close position to the Latch site. However, our current study clearly shows that similar to CDDO-Im and SFN, 15d-PGJ₂ does not use the Hinge-Latch mechanism.

As for the electrophilic inactivation of the Keap1 ubiquitin ligase activity, we propose that the cysteine modifications elicit structural alterations in Keap1 that do not bring about the dissociation of the Keap1–Nrf2 interaction, but brings about protein structural changes that consequently prevent Nrf2 ubiquitination. Supporting this proposal, conformational changes in Keap1 caused by Nrf2-activating compounds have been observed using a hydrophobicity probe[49]. Alternatively, it may also be possible that modification of Cys151 in the BTB domain affects the orientation angle of association between Keap1 and Nrf2, resulting in a change in the distance from ubiquitin to the target lysine residues in the Neh2 domain of Nrf2. Further structural analyses are required for revealing the electrophilic inactivation mechanism of Keap1.

In conclusion, this study has overcome the difficulties of simultaneous examination of two-site binding mechanism by use of titration NMR for competitive inhibition and demonstrates that the Hinge-Latch mechanism is actively utilized in the Nrf2 activation by the autophagy chaperone p62 and by pharmacological Keap1–Nrf2 PPI inhibitors, but not by electrophilic Nrf2 inducers. This study thus demonstrates that Keap1 utilizes multiple mechanisms for Nrf2 activation in response to a wide range of environmental stresses.

## Methods

**Chemical reagents.** 15-deoxy-$\Delta^{12,14}$-prostaglandin J₂ (15d-PGJ₂) and L-SFN were purchased from Cayman Chemical and Sigma Aldrich, respectively. 1-[2-cyano-3,12-dioxooleana-1,9(11)-dien-28-oyl] imidazole (CDDO-Im) was purchased from Namiki Co. Ltd. PRL295 and NG262 were synthesized in accordance with previously published protocols[34,35]. Non-phosphorylated p62-KIR peptide (KEVDPSTGELQSLQ) and phosphorylated p62-pKIR peptide (KEVDP-pS-TGELQSLQ) were purchased from Toray Research Center.

**Protein expression and purification.** Mouse Keap1 cDNA covering full-length Keap1 protein (Met1-Cys624) was inserted into the pET21a vector (Novagen) and C-terminally fused with 6×His tag. This protein was co-expressed in *Escherichia coli* BL21-Gold (DE3) (Novagen) with pG-Tf2 harboring chaperone genes by culturing in Terrific Broth medium at 37°C. Recombinant protein expression was induced by adding isopropyl-β-D-thiogalactopyranoside (IPTG; 0.2 mM). After overnight incubation at 15°C, the bacteria were harvested and mechanically lysed by sonication on ice (Branson Sonifier 450). The soluble-protein fraction was recovered by centrifugation at 10,000 rpm for 20 min at 4°C. After purification by affinity chromatography using Ni-NTA agarose (QIAGEN), Keap1 protein was further purified with Enrich Q 5 × 50 column (Bio-Rad) and HiLoad 16/60 Superdex 200 PG (Cytiva).

GST-TEV-mouse Nrf2-Neh2 cDNA was inserted into the pET15b vector (Novagen) and N-terminally fused with 6×His-GST tag. The protein was expressed in *E. coli* BL21-star (DE3) (Novagen) and the bacteria were cultured in a modified minimal M9 medium supplemented with $^{15}$N-ammonium chloride and $^{13}C_6$-D-glucose at 37°C. Expression of the recombinant protein was induced by adding IPTG (0.5 mM). After overnight incubation at 27°C, the cells were harvested and mechanically lysed by sonication on ice. The soluble-protein fraction was recovered by centrifugation at 10,000 rpm for 20 min at 4°C. After purification by affinity chromatography with Ni-NTA resin (QIAGEN), Neh2 with N-terminal 6×His-GST tag was cleaved by TurboTEV Protease (Accelagen) and the tag was removed by affinity chromatography using Ni-NTA resin. The protein was further purified with a series of column chromatography utilizing Enrich Q 5 × 50 (Bio-Rad) and Superdex 75 10/300GL (Cytiva).

**NMR data collection.** NMR sample of $^{13}C/^{15}N$-labeled Neh2 domain was prepared at a final concentration of 0.05 mM in 20 mM Tris-HCl buffer (pH 8.0), 100 mM NaCl, and 4 mM TCEP containing 10% $D_2O$. Two-dimensional TROSY-HSQC spectra[50] were acquired from the $^{13}C/^{15}N$-labeled Neh2 domain. All NMR experiments were performed at 298 K on a Bruker 800-MHz spectrometer equipped with a CryoProbe. The sequence-specific backbone $^1H^N$, $^{13}C^\alpha$, $^{13}C'$, and $^{15}N$ and side-chain $^{13}C^\beta$ resonance assignments of $^{13}C/^{15}N$-labeled Neh2 domain were performed by analyzing six 3D triple-resonance NMR spectra, CBCA(CO)NH, HNCACB, HNCA, HNCOCA, HNCO, and HN(CA)CO experiments[51,52]. All spectra were processed on a Linux PC by using the AZARA 2.7 software package (W. Boucher, www.bio.cam.ac.uk/azara). Spectra were analyzed on a Linux PC using CcpNmr Analysis version 2.4.0[53].

**NMR titration experiments.** For titration experiments of Keap1, 0.05-, and 0.1-mM unlabeled-Keap1 were added to the $^{13}C/^{15}N$-labeled Neh2 sample. For titration experiments with p62-KIR peptide, 0.05-mM (final) $^{13}C/^{15}N$-labeled Neh2, and 0.1-mM (final) Keap1 were dissolved in 20-mM Tris-HCl buffer (pH 8.0), 100-mM NaCl, and 4-mM TCEP, and the concentrated p62-KIR peptide was added to the sample. The final concentration of the peptide was 0.25-, 0.5-, 1.0-, 1.5-, and 5-mM (5-, 10-, 20-, 30-, and 100-fold of Neh2) for phosphorylated p62-KIR, and 1.0- and 5-mM (20- and 100-fold of Neh2) for non-phosphorylated p62-KIR. For titration experiments with PRL295, NG262, 15d-PGJ$_2$, CDDO-Im and SFN, 2D TROSY-HSQC spectra of $^{13}C/^{15}N$-labeled Neh2, and $^{13}C/^{15}N$-labeled Neh2–Keap1 complex were recorded in the presence of 5% DMSO-d6. Each compound was dissolved in DMSO-d6 at concentrated condition and added to the $^{13}C/^{15}N$-labeled Neh2–Keap1 complex. The final concentration of the solvent in the measured sample was 5%. The final concentrations of each compound were as follows; 0.025-, 0.05-, 0.1-, 0.15-, and 0.25-mM for both PRL295 and NG262, 0.05-, 0.25- and 0.55-mM for 15d-PGJ$_2$, 0.05-, 0.1-, and 0.5-mM for CDDO-Im, and 0.2- and 0.5-mM for SFN, respectively. For titration experiments with 15d-PGJ$_2$, CDDO-Im and SFN, 20-mM Tris-HCl (pH 8.0), 100-mM NaCl, and 1-mM TCEP were used as buffer. The relative intensities of the TROSY-HSQC signals of $^{13}C/^{15}N$-labeled Neh2 in the absence or presence of Keap1, p62-KIR peptide, and chemical compounds were plotted against the number of amino acids of Neh2. Analyses of the relative intensity changes were performed using CcpNmr Analysis version 2.4.0.

**Statistics and reproducibility.** A sufficient number of scans were used for all NMR experiments to obtain good signal-to-noise. All attempts to replicate the data were successful.

**Reporting summary.** Further information on research design is available in the Nature Research Reporting Summary linked to this article.

## Data availability
The source data for the graphs in the figures are available as Supplementary Data 1. All other data are available from the corresponding authors upon reasonable request.

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

## Acknowledgements

We would like to thank Dr. Yutaka Ito for technical support. This work was supported in part by MEXT/JSPS KAKENHI (19H05649 to M.Y., 17K07298 to S.K., and 19K07340 and 17KK0183 to T.S.), BINDS (JP21am0101095 to M.Y.) and P-CREATE (JP20cm0106101 to M.Y.) from AMED, and Takeda Science Foundation (M.Y. and T.S.). This work was also supported in part by the Tohoku Medical Megabank Project (JP19km0105001 and JP19km0105002), Project for Promoting Public Utilization of Advanced Research Infrastructure (MEXT), and Sharing and administrative network for research equipment (MEXT). We also thank the Biomedical Research Core of Tohoku University Graduate School of Medicine for technical support.

## Author contributions

T.S., T.M., A.D.K., T. Kamei, S.K., and M.Y. designed the research and analyzed the data. Y.H., T.S., J.I., A.D.K., S.K., and M.Y. wrote the manuscript. Y.H. and T.I. prepared the Keap1 and Neh2 proteins. J.I., T. Kasai, and S.K. conducted NMR experiment. G.W. and T.W.M. provided the Keap1–Nrf2 PPI inhibitors.

## Competing interests

The authors declare no competing interests.
