## [Peer Review File · Communications Biology]

Reviewers' comments:

Reviewer #1 (Remarks to the Author):

The Keap1-Nrf2 system is a critical regulator of cellular protective mechanisms. As it has protective effects in ageing-related degenerative diseases, pharmacological approaches to activate the system are highly sought for. The paper by Horie et al. examines the interaction of Keap1 and Nrf2 using a newly developed method where the binding of DLGex- and ETGE-binding motifs of Nrf2 to Keap1 is studied by NMR spectroscopy, using radiolabeled Neh2 domain of Nrf2 and full length Keap1. The authors conclude that the previously established hinge and latch model of Keap1-Nrf2 interaction holds true when protein-protein interaction (PPI) inhibitors are used to activate the system, whereas electrophilic inducers fail to trigger latch dissociation of DLGex-motif from Keap1.

Given the intense pharmacological development around the Keap1-Nrf2 system, the paper provides valuable information regarding the mechanism of action of different classes of Nrf2 activators. It is very clear from the data that PPI inhibitors elicit dissociation of Neh2 from Keap1, whereas electrophilic activators do not. Given that PP inhibitors NG262 or PRL295 used in the study are designed to bind to the pocket of Keap1 DC domain used also by DLGex and ETGE, the PPI inhibitor effect is to be expected, whereas the mechanism of action of electrophilic activators remain to be established. While the methodology used in the study does not allow addressing latter, it would be of interest if the authors would elaborate the mechanism a little further.

Reviewer #2 (Remarks to the Author):

This manuscript describes the interaction between the Neh2 domain of the transcription factor Nrf2 and the DC domain of Keap1. Neh2 contains two Keap1 binding motifs i.e. a low-affinity DLGex motif and a high-affinity ETGE motif. The authors have previously proposed the Hinge-Latch model, where the low-affinity motif first dissociates as a latch, while the ETGE motif remains bound as a hinge to Keap1. Several compounds have been shown to interfere with the Nrf2-Keap1 interaction including: 1) electrophilic compounds that react with cysteine thiols of Keap1, 2) disruptors that inhibit the Keap1-Nrf2 interaction as for example p62 that harbors a phosphorylated STGE motif with high affinity for the DC domain of Keap1 and 3) direct protein-protein interaction inhibitors such as PRL295 and NG262. In this manuscript, the authors investigate the effect of these three classes of compounds on the Neh2:Keap1-DC interaction using NMR titrations. The study provides valuable new insight, however, I am worried about the completeness of the NMR resonance assignments and the extent of potential exchange contributions to the line widths as outlined below:

1) The authors have previously used NMR to study the interaction mechanism of Nrf2-Neh2 with Keap1, however, as stated on line 107 the spectral assignments of Neh2 were limited at the time, justifying a new and more thorough study. Unfortunately, the NMR experiments in the current study have been carried out at pH 8 and 25C, where amide proton exchange rates are extremely fast leading to severe line broadening of the NMR resonances (in particular those of serine and threonine). For this reason, the spectral assignments in the current study remain incomplete compromising the entire study of the interaction mechanism. It is not clear to this reviewer exactly how many more assignments were obtained in the current study compared to the previous study, and more specifically why pH 8 was chosen as an experimental condition, when a complete assignment probably could have been obtained at pH 6.0 or even pH 7.0 (with NMR data acquisition at a lower temperature, for example 5C).

2) Line 141: The authors state: "However, to the best of our knowledge, this approach has not been applied for the analysis of protein-protein interactions involving two distinct sites." I am not sure what the authors mean exactly, but there are many cases in the literature where NMR titrations have been used to study binding of two sites in IDPs to one partner. See for example studies from Peter Wright, Julie Forman Kay & Martin Blackledge labs.

3) The interpretation of the line broadening observed in Neh2 upon addition of Keap1 does not

account for potential conformational exchange contributions to the NMR line widths. Caution should be taken when interpreting NMR intensity decreases in terms of binding affinities. Do the authors have additional evidence (for example from relaxation measurements at sub-stoichiometric ratios of Keap1) that conformational exchange contributions to the line widths can be ruled out?

4) Related to point 3), would it be possible that the different compounds tested modulate the interaction kinetics of the complex and therefore the line widths?

Reviewer #3 (Remarks to the Author):

The binding of the flexible Nrf2 NEH domain by the Keap1 E3 ligase is central to the cellular oxidative stress response and has been of significant interest to drug developers. Over many years, the authors and others have determined that Nrf2 contains two binding sites for Keap1, including a high affinity ETGE motif and a lower affinity DLGex motif. Since Keap1 is homodimeric, it has two substrate recognition domains that can bind simultaneously to both sites leading to a hinge (ETGE) and latch (DLGex) description. Disease mutations and previous work have suggested that both Nrf2 sites must engage Keap1 for efficient ubiquitination of Nrf2. This new study uses NMR titrations to probe these binding events.

The strengths of the work are (i) valuable peak assignments of 2D TROSY-HSQC spectra of Nrf2 NEH domain and experimental use of these through titration experiments; (ii) clear mechanistic hypotheses; and (iii) comparison of key reagents including different classes of chemical inhibitors, as well as the biological inhibitor protein (p62). There are, however, significant weaknesses in the manuscript. (i) the entire study only uses 1 experimental assay (NMR titration). Thus, predictions from NMR of intact Nrf2 binding in the presence of electrophiles are not replicated by cellular pull downs. Similarly, the inference of 1:2 stoichiometry is not validated by analytical ultracentrifugation. Cys modification is not confirmed by mass spec. (ii) the work merely confirms existing findings/models within the field and doesn't offer any breakthrough data. This is evident from the lack of orthogonal assays, which the authors likely considered are present in the historical literature. In summary, this is a nice to read technical piece of NMR work on an interesting biological system and more relevant for a specialised technical journal.

Other comments:

1) Overall, I like the final summing up paragraph (Page 20 lines 429-434) as it is a fair commentary. However, it is clear the insights from the manuscript are limited. For example, Page 13 line 273 "These findings provide valuable information for the development of next generation Nrf2 inducers that specifically inhibit the Keap1-Nrf2 interaction without electrophilic insults". However, the authors don't state what the valuable translatable knowledge is that is derived? Should inhibitors aim to only block DLGex?, or inhibit both sites?, or primarily aim to inhibit the Helix 2 interactions of DLGex? The study shows that it is easier to displace a weak affinity ligand (DLGex) than a higher affinity ligand (ETGE). Such rules are dictated by the laws of thermodynamics and don't need Nature Communications to confirm them.

2) A disadvantage of the NMR approach is that non-physiological concentrations are used e.g. 100 μ M KEAP1 concentration, 100-500 μ M small molecule inhibitors.

3) Page 13 Lines 282-288. It is stated that CDDO-Im was soluble up to 10x excess meaning 1 mM concentration. This is very surprising as it seems a highly insoluble molecule to work with. No mass spec is shown to prove the modification of Keap1 is near 100%. This mirrors the lack of orthogonal approaches anywhere in the manuscript.

Minor corrections:

- 1) The term Keap1-DC domain is used throughout. The wider community uses the term "Kelch domain". It would seem sensible to either use this nomenclature or mention "Kelch" domain once.
- 2) Page 10 line 217 typo "form" should be "from"
- 3) Page 11 line 233 proteins do not have a "top" and "bottom" or "left" and "right". The term "bottom" could be replaced.

- 4) Page 31 line 672 Figure 3 legend "Note that we quantitated monomeric Keap1 molecule, but not Keap1 homodimer molecule here." The word monomeric is inappropriate when discussing a dimer. Better to say you quantified it by concentration of the individual KEAP1 "protomers" rather than homodimer.
- 5) Figure 1A – why is there an arrow implying ubiquitination of Nrf2 leads only to DLGex dissociation as an outcome? If anything, shouldn't the arrow point the opposite direction i.e. once the DLGex binds ubiquitination can proceed?
- 6) Figure 1C top row NMR line broadening. I assume this is an entirely schematic depiction of a 1D NMR peak? Can the authors properly declare this in the legend as a schematic?
- 7) Figures 1d-g. These panels would normally be in supplemental if used in a high impact journal.
- 8) Figures 4a-e are poor quality – one cannot see the pocket; the compound is same colour as the side chains; there is no labelling. Panels in 4h and 4i are far better and actually sufficient (as our Supplemental Figure 2a-b).
- 9) Figures 4f-g. The key on the figure shows "0 PRL295; 1 PRL295; 2 PRL295". These are not defined in the legend. From the text I infer 1 means "equimolar with KEAP1-Nrf2 complex" and 2 means 2-fold excess over complex. Given the complex contains 2 KEAP1 protomers saturation of all KEAP1-DC sites is only expected with "2 PRL295". The authors should clarify the key in the legend.
- 10) Figure 7. Same argument as for Figure 1 regarding the direction of arrows after Nrf2 ubiquitination (as depicted). Ubiquitinated Nrf2 should only be directed to the proteasome not to stabilised products with compounds bound. Is it not better, to illustrate that the latch-dissociated states shown fail to be ubiquitinated? Similarly, it is suggested that electrophilic cmpds binding to Cys151 or Cys288 modify the IVR domain to perturb Cullin3 binding or stable positioning of the Keap1-DC (Kelch) domain. Could Cullin3 binding disruption be shown on the figure for these?

To Reviewer #1:

The Keap1-Nrf2 system is a critical regulator of cellular protective mechanisms. As it has protective effects in ageing-related degenerative diseases, pharmacological approaches to activate the system are highly sought for. The paper by Horie et al. examines the interaction of Keap1 and Nrf2 using a newly developed method where the binding of DLGex-and ETGE-binding motifs of Nrf2 to Keap1 is studied by NMR spectroscopy, using radiolabeled Neh2 domain of Nrf2 and full length Keap1. The authors conclude that the previously established hinge and latch model of Keap1-Nrf2 interaction holds true when protein-protein interaction (PPI) inhibitors are used to activate the system, whereas electrophilic inducers fail to trigger latch dissociation of DLGex-motif from Keap1.

Given the intense pharmacological development around the Keap1-Nrf2 system, the paper provides valuable information regarding the mechanism of action of different classes of Nrf2 activators. It is very clear from the data that PPI inhibitors elicit dissociation of Neh2 from Keap1, whereas electrophilic activators do not. Given that PPI inhibitors NG262 or PRL295 used in the study are designed to bind to the pocket of Keap1 DC domain used also by DLGex and ETGE, the PPI inhibitor effect is to be expected, whereas the mechanism of action of electrophilic activators remains to be established. While the methodology used in the study does not allow addressing latter, it would be of interest if the authors would elaborate the mechanism a little further.

We thank the reviewer for the professional comments. We believe that the electrophilic inactivation of Keap1 ubiquitin ligase activity is elicited by structural alterations in Keap1 without the dissociation of the Keap1-Nrf2 interaction.

Further structural analyses are required to answer the question, and we are planning to pursue the analyses in our next study. We have added a sentence explaining this point into Discussion (Page 19, line 24).

To Reviewer #2:

This manuscript describes the interaction between the Neh2 domain of the transcription factor Nrf2 and the DC domain of Keap1. Neh2 contains two Keap1 binding motifs, i.e., a low-affinity DLGex motif and a high-affinity ETGE motif. The authors have previously proposed the Hinge-Latch model, where the low-affinity motif first dissociates as a latch, while the ETGE motif remains bound as a hinge to Keap1. Several compounds have been shown to interfere with the Nrf2-Keap1 interaction including: 1) electrophilic compounds that react with cysteine thiols of Keap1, 2) disruptors that inhibit the Keap1-Nrf2 interaction as for example p62 that harbors a phosphorylated STGE motif with high affinity for the DC domain of Keap1 and 3) direct protein-protein interaction inhibitors such as PRL295 and NG262. In this manuscript, the authors investigate the effect of these three classes of compounds on the Neh2:Keap1-DC interaction using NMR titrations. The study provides valuable new insight; however, I am worried about the completeness of the NMR resonance assignments and the extent of potential exchange contributions to the line widths as outlined below:

We thank the reviewer for the professional comments. Whereas we have improved substantially in this study our previous assignment of the NMR resonance peaks (Tong et al, 2006), it is still a partial assignment as the reviewer pointed out. However, we would like to ask the reviewer to understand that it is very challenging to provide a complete assignment of the NMR resonance peaks in the case of Keap1-Nrf2 complex, as will be explained below, and we believe that it is beyond the scope of this paper.

1) The authors have previously used NMR to study the interaction mechanism of Nrf2-Neh2 with Keap1, however, as stated on line 107 the spectral assignments of Neh2 were limited at the time, justifying a new and more thorough study. Unfortunately, the NMR experiments in the current study have been carried out at pH 8 and 25C, where amide proton exchange rates are extremely fast leading to severe line broadening of the NMR resonances (in particular those of serine and threonine). For this reason, the spectral assignments in the current study remain incomplete compromising the entire study of the interaction mechanism. It is not clear to this reviewer exactly how many more assignments were obtained in the current study compared to the previous study, and more specifically why pH 8 was chosen as an experimental condition, when a complete assignment probably could have been obtained at pH 6.0 or even pH 7.0 (with NMR data acquisition at a lower temperature, for example 5C).

We agree with the reviewer that lower pH condition is better for the NMR experiments. However, in our repeated and accumulated trials for years, we found that the Keap1 dimer is unstable at pH less than 8.0 at protein concentrations suitable for the NMR experiments. This is true for not only us but for many leading laboratories in the world. Whereas we have improved substantially in this study our previous assignment of the NMR resonance peaks (Tong et al, 2006), it is still a partial assignment as the reviewer pointed out. However, it is technically not feasible at this point to provide a complete assignment of the NMR resonance peaks in the case of Keap1-Nrf2 complex.

On the other hand, whereas the quality of the spectra for the backbone assignment is

not the best or complete, we could clearly and sufficiently assign the important regions of Nrf2 for the interaction with Keap1, which is the critical question in the current Keap1-Nrf2 biology. Therefore, we have performed the NMR titration experiments at pH 8.0 in this study. We would like to ask the reviewer to understand the scope of our current study, that is we are pursuing to delineate whether the Hinge-Latch model is truly operating or not. We are not reporting the complete assignment of the NMR resonance peaks, which we feel is a totally distinct challenge. We have added an explanation for this point to the text (Page 9, line 12).

2) *Line 141: The authors state: “However, to the best of our knowledge, this approach has not been applied for the analysis of protein-protein interactions involving two distinct sites.” I am not sure what the authors mean exactly, but there are many cases in the literature where NMR titrations have been used to study binding of two sites in IDPs to one partner. See for example studies from Peter Wright, Julie Forman Kay & Martin Blackledge labs.*

We thank the reviewer for this professional advice. Based on the advice, we have extended our literature study more. We agree with the reviewer that there are reports, which show that the NMR titrations have been used for the study of protein-protein interactions involving two distinct sites. Therefore, we have deleted our previous description (Page 7, line 17).

3) *The interpretation of the line broadening observed in Neh2 upon addition of Keap1 does not account for potential conformational exchange contributions to the NMR line widths. Caution should be taken when interpreting NMR intensity decreases in terms of binding affinities. Do the authors have additional evidence (for example from relaxation measurements at sub-stoichiometric ratios of Keap1) that conformational exchange contributions to the line widths can be ruled out?*

We thank the reviewer for the professional and helpful comments. We agree with this point, but we do not have additional evidence to rule out conformational exchange contributions to the line widths. However, we surmise that the line broadening likely accounts for their interaction rather than conformational exchange, because co-crystal structures of Keap1-DLGex, Keap1-ETGE, and Keap1-PRL295 have shown that same pocket of Keap1 binds to DLGex, ETGE and PRL-295 (Padmanabhan et al, 2006; Fukutomi et al, 2014; Lazzara et al, 2020). We have added a sentence explaining this point (Page 7, line 12).

4) *Related to point 3), would it be possible that the different compounds tested modulate the interaction kinetics of the complex and therefore the line widths?*

We thank the reviewer for the interesting comments. The compounds tested in this study specifically bind to the pocket of Keap1, but not to Neh2. Therefore, we feel that it is unlikely that the compounds modulate the interaction kinetics of the complex.

To Reviewer #3:

The binding of the flexible Nrf2 NEH domain by the Keap1 E3 ligase is central to the cellular oxidative stress response and has been of significant interest to drug developers. Over many years, the authors and others have determined that Nrf2 contains two binding sites for Keap1, including a high affinity ETGE motif and a lower affinity DLGex motif. Since Keap1 is homodimeric, it has two substrate recognition domains that can bind simultaneously to both sites leading to a hinge (ETGE) and latch (DLGex) description. Disease mutations and previous work have suggested that both Nrf2 sites must engage Keap1 for efficient ubiquitination of Nrf2. This new study uses NMR titrations to probe these binding events.

The strengths of the work are (i) valuable peak assignments of 2D TROSY-HSQC spectra of Nrf2 NEH domain and experimental use of these through titration experiments; (ii) clear mechanistic hypotheses; and (iii) comparison of key reagents including different classes of chemical inhibitors, as well as the biological inhibitor protein (p62). There are, however, significant weaknesses in the manuscript. (i) the entire study only uses 1 experimental assay (NMR titration). Thus, predictions from NMR of intact Nrf2 binding in the presence of electrophiles are not replicated by cellular pull downs. Similarly, the inference of 1:2 stoichiometry is not validated by analytical ultracentrifugation. Cys modification is not confirmed by mass spec. (ii) the work merely confirms existing findings/models within the field and doesn't offer any breakthrough data. This is evident from the lack of orthogonal assays, which the authors likely considered are present in the historical literature. In summary, this is a nice to read technical piece of NMR work on an interesting biological system and more relevant for a specialised technical journal.

We apologize our lack of explanation of the preceding studies of pull downs, analytical ultracentrifugation and mass spec. There are many preceding reports of pull-down experiments, which reproducibly showed that electrophiles do not influence the Keap1-Nrf2 binding (Kobayashi et al, 2006; Zhang et al, 2004; Eggler 2005; Li et al, 2012). Similarly, we have already demonstrated unequivocally the 2:1 stoichiometry of Keap1-Nrf2 complex by using the analytical ultracentrifugation assay (Iso et al, 2016). As for the mass spec analyses, there are many studies by using mass spec including ours that showed that electrophiles SFN, 15d-PGJ2 and CDDO-Im are able to modify Keap1 (Hong et al, 2005; Kansanen et al, 2009; Kobayashi et al, 2009; Hu et al, 2011; Meng et al, 2020, and more). Therefore, we feel that repeating the suggested experiments in addition to the piles of these data, is unlikely to yield a meaningful outcome. We have added explanation and citation of previous pull-down studies (Page 4, line 25), analytical centrifugation (Page 4, line 14), mass spec (Page 5, line 8) in the revised manuscript.

Other comments:

1) Overall, I like the final summing up paragraph (Page 20 lines 429-434) as it is a fair commentary. However, it is clear the insights from the manuscript are limited. For example, Page 13 line 273 "These findings provide valuable information for the development of next generation Nrf2 inducers that specifically inhibit the Keap1-Nrf2

interaction without electrophilic insults”. However, the authors don’t state what the valuable translatable knowledge is that is derived? Should inhibitors aim to only block DLGex?, or inhibit both sites?, or primarily aim to inhibit the Helix 2 interactions of DLGex? The study shows that it is easier to displace a weak affinity ligand (DLGex) than a higher affinity ligand (ETGE). Such rules are dictated by the laws of thermodynamics and don’t need Nature Communications to confirm them.

This an excellent comment. Our primary goal of this study is to clarify the long-lasting scientific question as to whether the Hinge-Latch mechanism operates or not. We have succeeded in this point. On the contrary, as the development of PPI inhibitors for the Keap1-Nrf2 interaction is still in the dawn, one valuable translatable knowledge from the present accomplishments is that we can design specific chemicals, which attack the region in Keap1 pocket that interacts with Helix 2 of DLGex. As currently the design is based on the entire structure of the Keap1 pocket, current knowledge allows us to restrict the region in the pocket and to design more specific chemicals in rational drug design. We believe that this is an important support for the development of new generation PPI inhibiting drugs.

2) A disadvantage of the NMR approach is that non-physiological concentrations are used e.g. 100 μ M KEAP1 concentration, 100-500 μ M small molecule inhibitors.

We would like to ask the reviewer to understand that even with the limitations, still we are able to acquire important information, so that the current NMR titration approach is precious. We agree that other approaches are also valuable in order to approach for the physiological knowledge and we have been working on that direction many years.

3) Page 13 Lines 282-288. It is stated that CDDO-Im was soluble up to 10x excess meaning 1 mM concentration. This is very surprising as it seems a highly insoluble molecule to work with. No mass spec is shown to prove the modification of Keap1 is near 100%. This mirrors the lack of orthogonal approaches anywhere in the manuscript.

As reviewer commented, CDDO-Im is not so highly soluble, but we could solve it in 1 mM concentration and used for experiments. There have been many preceding papers that show that CDDO-Im can activate the Nrf2 pathway in vivo. Similarly, it has been shown by mass spec that CDDO-Im modifies Keap1 (Meng et al., 2020). We have cited the paper and added a sentence explaining this point (Page 5, line 8).

Minor corrections:

1) The term Keap1-DC domain is used throughout. The wider community uses the term “Kelch domain”. It would seem sensible to either use this nomenclature or mention “Kelch” domain once.

We thank the reviewer for this comment and have added description of “Kelch” to the text when it first appeared (Page 4, line 12).

2) Page 10 line 217 typo “form” should be “from”

We apologize for our oversight. We have corrected this error.

3) Page 11 line 233 proteins do not have a “top” and “bottom” or “left” and “right”. The term “bottom” could be replaced.

We have deleted the “bottom”.

4) Page 31 line 672 Figure 3 legend “Note that we quantitated monomeric Keap1 molecule, but not Keap1 homodimer molecule here.” The word monomeric is inappropriate when discussing a dimer. Better to say you quantified it by concentration of the individual KEAP1 “protomers” rather than homodimer.

We have corrected that “Note that we quantitated it by concentration of the individual Keap1 protomers rather than homodimer here”.

5) Figure 1A – why is there an arrow implying ubiquitination of Nrf2 leads only to DLGex dissociation as an outcome? If anything, shouldn't the arrow point the opposite direction i.e. once the DLGex binds ubiquitination can proceed?

We agree and have deleted the arrow.

6) Figure 1C top row NMR line broadening. I assume this is an entirely schematic depiction of a 1D NMR peak? Can the authors properly declare this in the legend as a schematic?

We have added phrase of schematic depiction of a 1D NMR peak in legend for Figure 1C (Page 31, line 12).

7) Figures 1d-g. These panels would normally be in supplemental if used in a high impact journal.

These panels show improvements accomplished firstly in this study. Whereas we previously utilized the DC domain protein that lacked the dimer-forming BTB domain of Keap1 (Tong et al, 2006), in this study we have succeeded in preparation of full-length Keap1 and obtained improved assignments. Therefore, we would like to retain the panels here.

8) Figures 4a-e are poor quality – one cannot see the pocket; the compound is same colour as the side chains; there is no labelling. Panels in 4h and 4i are far better and actually sufficient (as our Supplemental Figure 2a-b).

Following the advice, we have deleted Figure 4a-e.

9) Figures 4f-g. The key on the figure shows “0 PRL295; 1 PRL295; 2 PRL295”. These are not defined in the legend. From the text I infer 1 means “equimolar with KEAP1-Nrf2 complex” and 2 means 2-fold excess over complex. Given the complex contains 2 KEAP1

protomers saturation of all KEAP1-DC sites is only expected with “2 PRL295”. The authors should clarify the key in the legend.

We have added sentences explanation in figure legends (Page 32, line 22).

10) Figure 7. Same argument as for Figure 1 regarding the direction of arrows after Nrf2 ubiquitination (as depicted). Ubiquitinated Nrf2 should only be directed to the proteasome not to stabilised products with compounds bound. Is it not better, to illustrate that the latch-dissociated states shown fail to be ubiquitinated? Similarly, it is suggested that electrophilic cmpds binding to Cys151 or Cys288 modify the IVR domain to perturb Cullin3 binding or stable positioning of the Keap1-DC (Kelch) domain. Could Cullin3 binding disruption be shown on the figure for these?

The arrows here show that while in non-stimulus conditions Keap1-based E3 ligase ubiquitinates Nrf2, challenge of electrophilic inducers, PPI inhibitors or p62 brings about non-ubiquitinated Nrf2. Therefore, these arrows represent the challenges of Nrf2 inducers, and we believe they make sense.

We appreciate the reviewer for the latter professional comment. Disruption of the Cullin3 binding is likely, but still one of the possibilities and no solid evidence yet. Therefore, we would like to omit Cullin3 binding disruption in the Figure 7. This point needs to be studied rigorously.

REVIEWERS' COMMENTS:

Reviewer #1 (Remarks to the Author):

The authors have adequately addressed the reviewers' comments. I have no further questions.

Reviewer #2 (Remarks to the Author):

I am satisfied with the answers provided to my comments. I recommend publication of the manuscript.

Reviewer #3 (Remarks to the Author):

The authors' response is sufficient.